# Deep Learning with Logged Bandit Feedback

**Thorsten Joachims**
Cornell University
tj@cs.cornell.edu

**Adith Swaminathan**
Microsoft Research
adswamin@microsoft.com

**Maarten de Rijke**
University of Amsterdam
derijke@uva.nl

## Abstract

We propose a new output layer for deep neural networks that permits the use of logged contextual bandit feedback for training. Such contextual bandit feedback can be available in huge quantities (e.g., logs of search engines, recommender systems) at little cost, opening up a path for training deep networks on orders of magnitude more data. To this effect, we propose a counterfactual risk minimization approach for training deep networks using an equivariant empirical risk estimator with variance regularization, BanditNet, and show how the resulting objective can be decomposed in a way that allows stochastic gradient descent training. We empirically demonstrate the effectiveness of the method by showing how deep networks – ResNets in particular – can be trained for object recognition without conventionally labeled images.

## 1 Introduction

Log data can be recorded from online systems such as search engines, recommender systems, or online stores at little cost and in huge quantities. For concreteness, consider the interaction logs of an ad-placement system for banner ads. Such logs typically contain a record of the input to the system (e.g., features describing the user, banner ad, and page), the action that was taken by the system (e.g., a specific banner ad that was placed) and the feedback furnished by the user (e.g., clicks on the ad, or monetary payoff). This feedback, however, provides only partial information – "contextual-bandit feedback" – limited to the actions taken by the system. We do not get to see how the user would have responded, if the system had chosen a different action (e.g., other ads or banner types). Thus, the feedback for all other actions the system could have taken is typically not known. This makes learning from log data fundamentally different from traditional supervised learning, where "correct" predictions and a loss function provide feedback for all actions.

In this paper, we propose a new output layer for deep neural networks that allows training on logged contextual bandit feedback. By circumventing the need for full-information feedback, our approach opens a new and intriguing pathway for acquiring knowledge at unprecedented scale, giving deep neural networks access to this abundant and ubiquitous type of data. Similarly, it enables the application of deep learning even in domains where manually labeling full-information feedback is not viable.

In contrast to online learning with contextual bandit feedback (e.g., (Williams, 1992; Agarwal et al., 2014)), we perform *batch learning from bandit feedback* (BLBF) (Beygelzimer & Langford, 2009; Swaminathan & Joachims, 2015a;b;c) and the algorithm does not require the ability to make interactive interventions. At the core of the new output layer for BLBF training of deep neural networks lies a counterfactual training objective that replaces the conventional cross-entropy objective. Our approach – called BanditNet – follows the view of a deep neural network as a stochastic policy. We propose a counterfactual risk minimization (CRM) objective that is based on an equivariant estimator of the true error that only requires propensity-logged contextual bandit feedback. This makes our training objective fundamentally different from the conventional cross-entropy objective for supervised classification, which requires full-information feedback. Equivariance in our context means that the learning result is invariant to additive translations of the loss, and it is more formally defined in Section 3.2. To enable large-scale training, we show how this training objective can be decomposed to allow stochastic gradient descent (SGD) optimization.

In addition to the theoretical derivation of BanditNet, we present an empirical evaluation that verifies the applicability of the theoretical argument. It demonstrates how a deep neural network architec-

ture can be trained in the BLBF setting. In particular, we derive a BanditNet version of ResNet (He et al., 2016) for visual object classification. Despite using potentially much cheaper data, we find that Bandit-ResNet can achieve the same classification performance given sufficient amounts of contextual bandit feedback as ResNet trained with cross-entropy on conventionally (full-information) annotated images. To easily enable experimentation on other applications, we share an implementation of BanditNet.[1]

## 2 RELATED WORK

Several recent works have studied weak supervision approaches for deep learning. Weak supervision has been used to pre-train good image features (Joulin et al., 2016) and for information retrieval (Dehghani et al., 2017). Closely related works have studied label corruption on CIFAR-10 recently (Zhang et al., 2016). However, all these approaches use weak supervision/corruption to construct noisy proxies for labels, and proceed with traditional supervised training (using cross-entropy or mean-squared-error loss) with these proxies. In contrast, we work in the BLBF setting, which is an orthogonal data-source, and modify the loss functions optimized by deep nets to directly implement risk minimization.

Virtually all previous methods that can learn from logged bandit feedback employ some form of risk minimization principle (Vapnik, 1998) over a model class. Most of the methods (Beygelzimer & Langford, 2009; Bottou et al., 2013; Swaminathan & Joachims, 2015a) employ an inverse propensity scoring (IPS) estimator (Rosenbaum & Rubin, 1983) as empirical risk and use stochastic gradient descent (SGD) to optimize the estimate over large datasets. Recently, the self-normalized estimator (Trotter & Tukey, 1956) has been shown to be a more suitable estimator for BLBF (Swaminathan & Joachims, 2015c). The self-normalized estimator, however, is not amenable to stochastic optimization and scales poorly with dataset size. In our work, we demonstrate how we can efficiently optimize a reformulation of the self-normalized estimator using SGD.

Previous BLBF methods focus on simple model classes: log-linear and exponential models (Swaminathan & Joachims, 2015a) or tree-based reductions (Beygelzimer & Langford, 2009). In contrast, we demonstrate how current deep learning models can be trained effectively via batch learning from bandit feedback (BLBF), and compare these with existing approaches on a benchmark dataset (Krizhevsky & Hinton, 2009).

Our work, together with independent concurrent work (Serban et al., 2017), demonstrates success with *off-policy* variants of the REINFORCE (Williams, 1992) algorithm. In particular, our algorithm employs a Lagrangian reformulation of the self-normalized estimator, and the objective and gradients of this reformulation are similar in spirit to the updates of the REINFORCE algorithm. This connection sheds new light on the role of the baseline hyper-parameters in REINFORCE: rather than simply reduce the variance of policy gradients, our work proposes a constructive algorithm for selecting the baseline in the off-policy setting and it suggests that the baseline is instrumental in creating an equivariant counterfactual learning objective.

## 3 BANDITNET: COUNTERFACTUAL RISK MINIMIZATION FOR DEEP NETS

To formalize the problem of batch learning from bandit feedback for deep neural networks, consider the contextual bandit setting where a policy $\pi$ takes as input $x \in \mathcal{X}$ and outputs an action $y \in \mathcal{Y}$. In response, we observe the loss (or payoff) $\delta(x, y)$ of the selected action $y$, where $\delta(x, y)$ is an arbitrary (unknown) function that maps actions and contexts to a bounded real number. For example, in display advertising, the context $x$ could be a representation of the user and page, $y$ denotes the displayed ad, and $\delta(x, y)$ could be the monetary payoff from placing the ad (zero if no click, or dollar amount if clicked). The contexts are drawn i.i.d. from a fixed but unknown distribution $\Pr(X)$.

In this paper, a (deep) neural network is viewed as implementing a stochastic policy $\pi$. We can think of such a network policy as a conditional distribution $\pi_w(Y \mid x)$ over actions $y \in Y$, where $w$ are the parameters of the network. The network makes a prediction by sampling an action $y \sim \pi_w(Y \mid x)$, where deterministic $\pi_w(Y \mid x)$ are a special case. As we will show as part of the empirical

---

[1] http://www.joachims.org/banditnet/

evaluation, many existing network architectures are compatible with this stochastic-policy view. For example, any network $f_w(x, y)$ with a softmax output layer

$$\pi_w(y \mid x) = \frac{\exp(f_w(x, y))}{\sum_{y' \in \mathcal{Y}} \exp(f_w(x, y'))} \tag{1}$$

can be re-purposed as a conditional distribution from which one can sample actions, instead of interpreting it as a conditional likelihood like in full-information supervised learning.

The goal of learning is to find a policy $\pi_w$ that minimizes the risk (analogously: maximizes the payoff) defined as

$$R(\pi_w) = \mathop{\mathbb{E}}_{x \sim \Pr(X)} \mathop{\mathbb{E}}_{y \sim \pi_w(Y|x)} [\delta(x, y)]. \tag{2}$$

Any data collected from an interactive system depends on the policy $\pi_0$ that was running on the system at the time, determining which actions $y$ and losses $\delta(x, y)$ are observed. We call $\pi_0$ the *logging policy*, and for simplicity assume that it is stationary. The logged data $D$ are $n$ tuples of observed context $x_i \sim \Pr(X)$, action $y_i \sim \pi_0(Y \mid x_i)$ taken by the logging policy, the probability of this action $p_i \equiv \pi_0(y_i \mid x_i)$, which we call the *propensity*, and the received loss $\delta_i \equiv \delta(x_i, y_i)$:

$$D = [(x_1, y_1, p_1, \delta_1), \ldots, (x_n, y_n, p_n, \delta_n)]. \tag{3}$$

We will now discuss how we can use this logged contextual bandit feedback to train a neural network policy $\pi_w(Y \mid x)$ that has low risk $R(\pi_w)$.

## 3.1 COUNTERFACTUAL RISK MINIMIZATION

While conditional maximum likelihood is a standard approach for training deep neural networks, it requires that the loss $\delta(x_i, y)$ is known for all $y \in \mathcal{Y}$. However, we only know $\delta(x_i, y_i)$ for the particular $y_i$ chosen by the logging policy $\pi_0$. We therefore take a different approach following (Langford et al., 2008; Swaminathan & Joachims, 2015b), where we directly minimize an empirical risk that can be estimated from the logged bandit data $D$. This approach is called *counterfactual risk minimization* (CRM) (Swaminathan & Joachims, 2015b), since for any policy $\pi_w$ it addresses the counterfactual question of how well that policy would have performed, if it had been used instead of $\pi_0$.

While minimizing an empirical risk as an estimate of the true risk $R(\pi_w)$ is a common principle in machine learning (Vapnik, 1998), getting a reliable estimate based on the training data $D$ produced by $\pi_0$ is not straightforward. The logged bandit data $D$ is not only incomplete (i.e., we lack knowledge of $\delta(x_i, y)$ for many $y \in \mathcal{Y}$ that $\pi_w$ would have chosen differently from $\pi_0$), but it is also biased (i.e., the actions preferred by $\pi_0$ are over-represented). This is why existing work on training deep neural networks either requires full knowledge of the loss function, or requires the ability to interactively draw new samples $y_i \sim \pi_w(Y \mid x_i)$ for any new policy $\pi_w$. In our setting we can do neither – we have a fixed dataset $D$ that is limited to samples from $\pi_0$.

To nevertheless get a useful estimate of the empirical risk, we explicitly address both the bias and the variance of the risk estimate. To correct for sampling bias and handle missing data, we approach the risk estimation problem using importance sampling and thus remove the distribution mismatch between $\pi_0$ and $\pi_w$ (Langford et al., 2008; Owen, 2013; Swaminathan & Joachims, 2015b):

$$R(\pi_w) = \mathop{\mathbb{E}}_{x \sim \Pr(X)} \mathop{\mathbb{E}}_{y \sim \pi_w(Y|x)} [\delta(x, y)] = \mathop{\mathbb{E}}_{x \sim \Pr(X)} \mathop{\mathbb{E}}_{y \sim \pi_0(Y|x)} \left[ \delta(x, y) \frac{\pi_w(y \mid x)}{\pi_0(y \mid x)} \right]. \tag{4}$$

The latter expectation can be estimated on a sample $D$ of $n$ bandit-feedback examples using the following IPS estimator (Langford et al., 2008; Owen, 2013; Swaminathan & Joachims, 2015b):

$$\hat{R}_{IPS}(\pi_w) = \frac{1}{n} \sum_{i=1}^{n} \delta_i \frac{\pi_w(y_i \mid x_i)}{\pi_0(y_i \mid x_i)}. \tag{5}$$

This IPS estimator is unbiased and has bounded variance, if the logging policy has full support in the sense that $\forall x, y : \pi_0(y \mid x) \geq \epsilon > 0$. While at first glance it may seem natural to directly train the parameters $w$ of a network to optimize this IPS estimate as an empirical risk, there are at least three obstacles to overcome. First, we will argue in the following section that the naive IPS

estimator's lack of equivariance makes it sub-optimal for use as an empirical risk for high-capacity models. Second, we have to find an efficient algorithm for minimizing the empirical risk, especially making it accessible to stochastic gradient descent (SGD) optimization. And, finally, we are faced with an unusual type of bias-variance trade-off since "distance" from the exploration policy impacts the variance of the empirical risk estimate for different $w$.

## 3.2 EQUIVARIANT COUNTERFACTUAL RISK MINIMIZATION

While Eq. (5) provides an unbiased empirical risk estimate, it exhibits the – possibly severe – problem of "propensity overfitting" when directly optimized within a learning algorithm (Swaminathan & Joachims, 2015c). It is a problem of overfitting to the choices $y_i$ of the logging policy, and it occurs on top of the normal overfitting to the $\delta_i$. Propensity overfitting is linked to the lack of equivariance of the IPS estimator: while the minimizer of true risk $R(\pi_w)$ does not change when translating the loss by a constant (i.e., $\forall x, y : \delta(x, y) + c$) by linearity of expectation,

$$c + \min_w \mathop{\mathbb{E}}_{x \sim \Pr(X)} \mathop{\mathbb{E}}_{y \sim \pi_w(Y|x)} [\delta(x,y)] = \min_w \mathop{\mathbb{E}}_{x \sim \Pr(X)} \mathop{\mathbb{E}}_{y \sim \pi_w(Y|x)} [\delta(x,y) + c] \qquad (6)$$

the minimizer of the IPS-estimated empirical risk $\hat{R}_{IPS}(\pi_w)$ can change dramatically for finite training samples, and

$$c + \min_w \frac{1}{n} \sum_{i=1}^n \delta_i \frac{\pi_w(y_i \mid x_i)}{\pi_0(y_i \mid x_i)} \neq \min_w \frac{1}{n} \sum_{i=1}^n (\delta_i + c) \frac{\pi_w(y_i \mid x_i)}{\pi_0(y_i \mid x_i)}. \qquad (7)$$

Intuitively, when $c$ shifts losses to be positive numbers, policies $\pi_w$ that put as little probability mass as possible on the observed actions have low risk estimates. If $c$ shifts the losses to the negative range, the exact opposite is the case. For either choice of $c$, the choice of the policy eventually selected by the learning algorithm can be dominated by where $\pi_0$ happens to sample data, not by which actions have low loss.

The following self-normalized IPS estimator (SNIPS) addresses the propensity overfitting problem (Swaminathan & Joachims, 2015c) and is equivariant:

$$\hat{R}_{SNIPS}(\pi_w) = \frac{\frac{1}{n} \sum_{i=1}^n \delta_i \frac{\pi_w(y_i|x_i)}{\pi_0(y_i|x_i)}}{\frac{1}{n} \sum_{i=1}^n \frac{\pi_w(y_i|x_i)}{\pi_0(y_i|x_i)}}. \qquad (8)$$

In addition to being equivariant, this estimate can also have substantially lower variance than Eq. (5), since it exploits the knowledge that the denominator

$$S := \frac{1}{n} \sum_{i=1}^n \frac{\pi_w(y_i \mid x_i)}{\pi_0(y_i \mid x_i)} \qquad (9)$$

always has expectation 1:

$$\mathbb{E}[S] = \frac{1}{n} \sum_{i=1}^n \int \frac{\pi_w(y_i \mid x_i)}{\pi_0(y_i \mid x_i)} \pi_0(y_i \mid x_i) \Pr(x_i) dy_i dx_i = \frac{1}{n} \sum_{i=1}^n \int 1 \Pr(x_i) dx_i = 1. \qquad (10)$$

The SNIPS estimator uses this knowledge as a multiplicative control variate (Swaminathan & Joachims, 2015c). While the SNIPS estimator has some bias, this bias asymptotically vanishes at a rate of $O(\frac{1}{n})$ (Hesterberg, 1995). Using the SNIPS estimator as our empirical risk implies that we need to solve the following optimization problem for training:

$$\hat{w} = \operatorname*{arg\,min}_{w \in \Re^N} \hat{R}_{SNIPS}(\pi_w). \qquad (11)$$

Thus, we now turn to designing efficient optimization methods for this training objective.

## 3.3 TRAINING ALGORITHM

Unfortunately, the training objective in Eq. (11) does not permit stochastic gradient descent (SGD) optimization in the given form (see Appendix C), which presents an obstacle to efficient and effective training of the network. To remedy this problem, we will now develop a reformulation that retains

both the desirable properties of the SNIPS estimator, as well as the ability to reuse established SGD training algorithms. Instead of optimizing a ratio as in Eq. (11), we will reformulate the problem into a series of constrained optimization problems. Let $\hat{w}$ be a solution of Eq. (11), and at that solution let $S^*$ be the value of the control variate for $\pi_{\hat{w}}$ as defined in Eq. (9). For simplicity, assume that the minimizer $\hat{w}$ is unique. If we knew $S^*$, we could equivalently solve the following constrained optimization problem:

$$\hat{w} = \underset{w \in \Re^N}{\arg\min} \frac{1}{n} \sum_{i=1}^n \delta_i \frac{\pi_w(y_i \mid x_i)}{\pi_0(y_i \mid x_i)} \quad \text{subject to} \quad \frac{1}{n} \sum_{i=1}^n \frac{\pi_w(y_i \mid x_i)}{\pi_0(y_i \mid x_i)} = S^*. \tag{12}$$

Of course, we do not actually know $S^*$. However, we can do a grid search in $\{S_1, \ldots, S_k\}$ for $S^*$ and solve the above optimization problem for each value, giving us a set of solutions $\{\hat{w}_1, \ldots, \hat{w}_k\}$. Note that $S$ is just a one-dimensional quantity, and that the sensible range we need to search for $S^*$ concentrates around 1 as $n$ increases (see Appendix B). To find the overall (approximate) $\hat{w}$ that optimizes the SNIPS estimate, we then simply take the minimum:

$$\hat{w} = \underset{(\hat{w}_j, S_j)}{\arg\min} \frac{\frac{1}{n} \sum_{i=1}^n \delta_i \frac{\pi_{\hat{w}_j}(y_i \mid x_i)}{\pi_0(y_i \mid x_i)}}{S_j}. \tag{13}$$

This still leaves the question of how to solve each equality constrained risk minimization problem using SGD. Fortunately, we can perform an equivalent search for $S^*$ without constrained optimization. To this effect, consider the Lagrangian of the constrained optimization problem in Eq. (12) with $S_j$ in the constraint instead of $S^*$:

$$L(w, \lambda) = \frac{1}{n} \sum_{i=1}^n \frac{\delta_i \pi_w(y_i \mid x_i)}{\pi_0(y_i \mid x_i)} - \lambda \left[ \frac{1}{n} \left( \sum_{i=1}^n \frac{\pi_w(y_i \mid x_i)}{\pi_0(y_i \mid x_i)} \right) - S_j \right] = \frac{1}{n} \sum_{i=1}^n \frac{(\delta_i - \lambda) \pi_w(y_i \mid x_i)}{\pi_0(y_i \mid x_i)} + \lambda S_j.$$

The variable $\lambda$ is an unconstrained Lagrange multiplier. To find the minimum of Eq. (12) for a particular $S_j$, we need to minimize $L(w, \lambda)$ w.r.t. $w$ and maximize w.r.t. $\lambda$.

$$\hat{w}_j = \underset{w \in \Re^N}{\arg\min} \max_\lambda L(w, \lambda) \tag{14}$$

However, we are not actually interested in the constrained solution of Eq. (12) for any specific $S_j$. We are merely interested in exploring a certain range $S \in [S_1, S_k]$ in our search for $S^*$. So, we can reverse the roles of $\lambda$ and $S$, where we keep $\lambda$ fixed and determine the corresponding $S$ in hindsight. In particular, for each $\{\lambda_1, \ldots, \lambda_k\}$ we solve

$$\hat{w}_j = \underset{w \in \Re^N}{\arg\min} L(w, \lambda_j). \tag{15}$$

Note that the solution $\hat{w}_j$ does not depend on $S_j$, so we can compute $S_j$ after we have found the minimum $\hat{w}_j$. In particular, we can determine the $S_j$ that corresponds to the given $\lambda_j$ using the necessary optimality conditions,

$$\frac{\partial L}{\partial w} = \frac{1}{n} \sum_{i=1}^n \frac{\partial \pi_w(y_i \mid x_i)}{\partial w} \frac{(\delta_i - \lambda_j)}{\pi_0(y_i \mid x_i)} = 0 \quad \text{and} \quad \frac{\partial L}{\partial \lambda_j} = \frac{1}{n} \sum_{i=1}^n \frac{\pi_w(y_i \mid x_i)}{\pi_0(y_i \mid x_i)} - S_j = 0, \tag{16}$$

by solving the second equality of Eq. (16). In this way, the sequence of $\lambda_j$ produces solutions $\hat{w}_j$ corresponding to a sequence of $\{S_1, \ldots, S_k\}$.

To identify the sensible range of $S$ to explore, we can make use of the fact that Eq. (9) concentrates around its expectation of 1 for each $\pi_w$ as $n$ increases. Theorem 2 in Appendix B provides a characterization of how large the range needs to be. Furthermore, we can steer the exploration of $S$ via $\lambda$, since the resulting $S$ changes monotonically with $\lambda$:

$$(\lambda_a < \lambda_b) \text{ and } (\hat{w}_a \neq \hat{w}_b \text{ are not equivalent optima in Eq. (15))} \Rightarrow (S_a < S_b). \tag{17}$$

A more formal statement and proof are given as Theorem 1 in Appendix A. In the simplest form one could therefore perform a grid search on $\lambda$, but more sophisticated search methods are possible too.

After this reformulation, the key computational problem is finding the solution of Eq. (15) for each $\lambda_j$. Note that in this unconstrained optimization problem, the Lagrange multiplier effectively translates the loss values in the conventional IPS estimate:

$$\hat{w}_j = \underset{w}{\arg\min} \frac{1}{n} \sum_{i=1}^n (\delta_i - \lambda_j) \frac{\pi_w(y_i \mid x_i)}{\pi_0(y_i \mid x_i)} = \underset{w}{\arg\min} \hat{R}_{IPS}^{\lambda_j}(\pi_w). \tag{18}$$

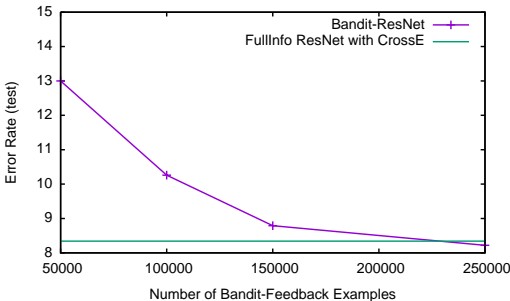

Figure 1: Learning curve of BanditNet. The x-axis is the amount of bandit feedback, the y-axis is the test error. Given enough bandit feedback, Bandit-ResNet converges to the skyline performance.

We denote this $\lambda$-translated IPS estimate with $\hat{R}_{IPS}^{\lambda}(\pi_w)$. Note that each such optimization problem is now in the form required for SGD, where we merely weight the derivative of the stochastic policy network $\pi_w(y \mid x)$ by a factor $(\delta_i - \lambda_j)/\pi_0(y_i \mid x_i)$. This opens the door for re-purposing existing fast methods for training deep neural networks, and we demonstrate experimentally that SGD with momentum is able to optimize our objective scalably.

Similar loss translations have previously been used in on-policy reinforcement learning (Williams, 1992), where they are motivated as minimizing the variance of the gradient estimate (Weaver & Tao, 2001; Greensmith et al., 2004). However, the situation is different in the off-policy setting we consider. First, we cannot sample new roll-outs from the current policy under consideration, which means we cannot use the standard variance-optimal estimator used in REINFORCE. Second, we tried using the (estimated) expected loss of the learned policy as the baseline as is commonly done in REINFORCE, but will see in the experiment section that this value for $\lambda$ is far from optimal. Finally, it is unclear whether gradient variance, as opposed to variance of the ERM objective, is really the key issue in batch learning from bandit feedback. In this sense, our approach provides a rigorous justification and a constructive way of picking the value of $\lambda$ in the off-policy setting – namely the value for which the corresponding $S_j$ minimizes Eq. (13). In addition, one can further add variance regularization (Swaminathan & Joachims, 2015b) to improve the robustness of the risk estimate in Eq. (18) (see Appendix D for details).

## 4    EMPIRICAL EVALUATION

The empirical evaluation is designed to address three key questions. First, it verifies that deep models can indeed be trained effectively using our approach. Second, we will compare how the same deep neural network architecture performs under different types of data and training objectives – in particular, conventional cross-entropy training using full-information data. In order to be able to do this comparison, we focus on synthetic contextual bandit feedback data for training BanditNet that is sampled from the full-information labels. Third, we explore the effectiveness and fidelity of the approximate SNIPS objective.

For the following BanditNet experiments, we adapted the ResNet20 architecture (He et al., 2016) by replacing the conventional cross-entropy objective with our counterfactual risk minimization objective. We evaluate the performance of this Bandit-ResNet on the CIFAR-10 (Krizhevsky & Hinton, 2009) dataset, where we can compare training on full-information data with training on bandit feedback, and where there is a full-information test set for estimating prediction error.

To simulate logged bandit feedback, we perform the standard supervised to bandit conversion (Beygelzimer & Langford, 2009). We use a hand-coded logging policy that achieves about 49% error rate on the training data, which is substantially worse than what we hope to achieve after learning. This emulates a real world scenario where one would bootstrap an operational system with a mediocre policy (e.g., derived from a small hand-labeled dataset) and then deploys it to log bandit feedback. This logged bandit feedback data is then used to train the Bandit-ResNet.

We evaluate the trained model using error rate on the held out (full-information) test set. We compare this model against the skyline of training a conventional ResNet using the full-information feedback

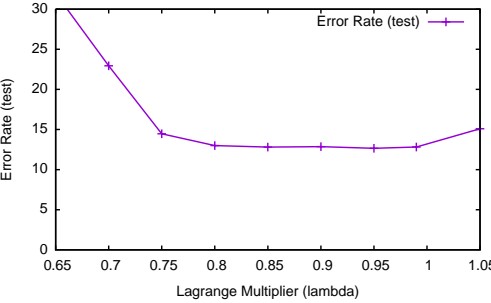 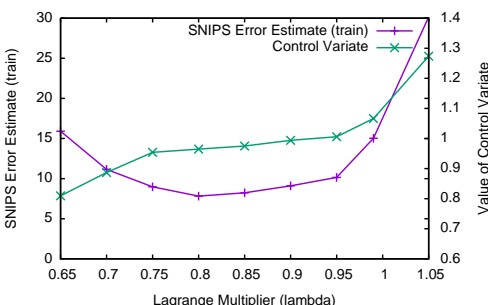

Figure 2: The x-axis shows the value of the Lagrange multiplier $\lambda$ used for training. Left plot shows the test error. Right plot shows the value of the SNIPS objective and the normalizer $S$. The size of the training set is 50k bandit-feedback examples.

from the 50,000 training examples. Both the conventional full-information ResNet as well as the Bandit-ResNet use the same network architecture, the same hyperparameters, the same data augmentation scheme, and the same optimization method that were set in the CNTK implementation of ResNet20. Since CIFAR10 does not come with a validation set for tuning the variance-regularization constant $\gamma$, we do not use variance regularization for Bandit-ResNet. The Lagrange multiplier $\lambda \in \{0.65, 0.7, 0.75, 0.8, 0.85, 0.9, 0.95, 1.0, 1.05\}$ is selected on the training set via Eq. (13). The only parameter we adjusted for Bandit-ResNet is lowering the learning rate to $0.1$ and slowing down the learning rate schedule. The latter was done to avoid confounding the Bandit-ResNet results with potential effects from early stopping, and we report test performance after 1000 training epochs, which is well beyond the point of convergence in all runs.

**Learning curve.** Figure 1 shows the prediction error of the Bandit-ResNet as more and more bandit feedback is provided for training. First, even though the logging policy that generated the bandit feedback has an error rate of 49%, the prediction error of the policy learned by the Bandit-ResNet is substantially better. It is between 13% and 8.2%, depending on the amount of training data. Second, the horizontal line is the performance of a conventional ResNet trained on the full-information training set. It serves as a skyline of how good Bandit-ResNet could possibly get given that it is sampling bandit feedback from the same full-information training set. The learning curve in Figure 1 shows that Bandit-ResNet converges to the skyline performance given enough bandit feedback training data, providing strong evidence that our training objective and method can effectively extract the available information provided in the bandit feedback.

**Effect of the choice of Lagrange multiplier.** The left-hand plot in Figure 2 shows the test error of solutions $\hat{w}_j$ depending on the value of the Lagrange multiplier $\lambda_j$ used during training. It shows that $\lambda$ in the range $0.8$ to $1.0$ results in good prediction performance, but that performance degrades outside this area. The SNIPS estimates in the right-hand plot of Figure 2 roughly reflects this optimal range, given empirical support for both the SNIPS estimator and the use of Eq. (13).

We also explored two other methods for selecting $\lambda$. First, we used the straightforward IPS estimator as the objective (i.e., $\lambda = 0$), which leads to prediction performance worse than that of the logging policy (not shown). Second, we tried using the (estimated) expected loss of the learned policy as the baseline as is commonly done in REINFORCE. As Figure 1 shows, it is between $0.130$ and $0.083$ for the best policies we found. Figure 2 (left) shows that these baseline values are well outside of the optimum range.

Also shown in the right-hand plot of Figure 2 is the value of the control variate in the denominator of the SNIPS estimate. As expected, it increases from below $1$ to above $1$ as $\lambda$ is increased. Note that large deviations of the control variate from $1$ are a sign of propensity overfitting (Swaminathan & Joachims, 2015c). In particular, for all solutions $\hat{w}_j$ the estimated standard error of the control variate $S_j$ was less than $0.013$, meaning that the normal 95% confidence interval for each $S_j$ is contained in $[0.974, 1.026]$. If we see a $\hat{w}_j$ with control variate $S_j$ outside this range, we should be suspicious of propensity overfitting to the choices of the logging policy and discard this solution.

## 5 CONCLUSIONS AND FUTURE WORK

We proposed a new output layer for deep neural networks that enables the use of logged contextual bandit feedback for training. This type of feedback is abundant and ubiquitous in the form of interaction logs from autonomous systems, opening up the possibility of training deep neural networks on unprecedented amounts of data. In principle, this new output layer can replace the conventional cross-entropy layer for any network architecture. We provide a rigorous derivation of the training objective, linking it to an equivariant counterfactual risk estimator that enables counterfactual risk minimization. Most importantly, we show how the resulting training objective can be decomposed and reformulated to make it feasible for SGD training. We find that the BanditNet approach applied to the ResNet architecture achieves predictive accuracy comparable to conventional full-information training for visual object recognition.

The paper opens up several directions for future work. First, it enables many new applications where contextual bandit feedback is readily available. Second, in settings where it is infeasible to log propensity-scored data, it would be interesting to combine BanditNet with propensity estimation techniques. Third, there may be improvements to BanditNet, like smarter search techniques for $S$, more efficient counterfactual estimators beyond SNIPS, and the ability to handle continuous outputs.

### ACKNOWLEDGMENTS

This research was supported in part by NSF Award IIS-1615706, a gift from Bloomberg, the Criteo Faculty Research Award program, and the Netherlands Organisation for Scientific Research (NWO) under project nr. 612.001.116. All content represents the opinion of the authors, which is not necessarily shared or endorsed by their respective employers and/or sponsors.

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

## A  APPENDIX: STEERING THE EXPLORATION OF $S$ THROUGH $\lambda$.

**Theorem 1.** *Let $\lambda_a < \lambda_b$ and let*

$$\hat{w}_a = \arg\min_w \hat{R}_{IPS}^{\lambda_a}(\pi_w) \tag{19}$$

$$\hat{w}_b = \arg\min_w \hat{R}_{IPS}^{\lambda_b}(\pi_w). \tag{20}$$

*If the optima $\hat{w}_a$ and $\hat{w}_b$ are not equivalent in the sense that $\hat{R}_{IPS}^{\lambda_a}(\pi_{\hat{w}_a}) \neq \hat{R}_{IPS}^{\lambda_a}(\pi_{\hat{w}_b})$ and $\hat{R}_{IPS}^{\lambda_b}(\pi_{\hat{w}_a}) \neq \hat{R}_{IPS}^{\lambda_b}(\pi_{\hat{w}_b})$, then*

$$S_a < S_b. \tag{21}$$

*Proof.* Abbreviate $f(w) = \frac{1}{n}\sum_{i=1}^n \delta_i \frac{\pi_w(y_i|x_i)}{\pi_0(y_i|x_i)}$ and $g(w) = \frac{1}{n}\sum_{i=1}^n \frac{\pi_w(y_i|x_i)}{\pi_0(y_i|x_i)}$. Then

$$\hat{R}_{IPS}^{\lambda}(\pi_w) = f(w) - \lambda g(w), \tag{22}$$

where $g(w)$ corresponds to the value of the control variate $S$. Since $\hat{w}_a$ and $\hat{w}_b$ are not equivalent optima, we know that

$$f(\hat{w}_a) - \lambda_a\, g(\hat{w}_a) \quad < \quad f(\hat{w}_b) - \lambda_a\, g(\hat{w}_b) \tag{23}$$

$$f(\hat{w}_b) - \lambda_b\, g(\hat{w}_b) \quad < \quad f(\hat{w}_a) - \lambda_b\, g(\hat{w}_a) \tag{24}$$

Adding the two inequalities and solving implies that

$$\Rightarrow \quad f(\hat{w}_a) - \lambda_a\, g(\hat{w}_a) + f(\hat{w}_b) - \lambda_b\, g(\hat{w}_b) < f(\hat{w}_b) - \lambda_a\, g(\hat{w}_b) + f(\hat{w}_a) - \lambda_b\, g(\hat{w}_a) \tag{25}$$

$$\Leftrightarrow \quad \lambda_a\, g(\hat{w}_a) + \lambda_b\, g(\hat{w}_b) > \lambda_a\, g(\hat{w}_b) + \lambda_b\, g(\hat{w}_a) \tag{26}$$

$$\Leftrightarrow \quad (\lambda_b - \lambda_a)\, g(\hat{w}_b) > (\lambda_b - \lambda_a)\, g(\hat{w}_a) \tag{27}$$

$$\Leftrightarrow \quad g(\hat{w}_b) > g(\hat{w}_a) \tag{28}$$

$$\Leftrightarrow \quad S_b > S_a \quad \square \tag{29}$$

## B  APPENDIX: CHARACTERIZING THE RANGE OF $S$ TO EXPLORE.

**Theorem 2.** *Let $p \leq \pi_0(y \mid x)$ be a lower bound on the propensity for the logging policy, then constraining the solution of Eq. (11) to the $w$ with control variate $S \in [1 - \epsilon, 1 + \epsilon]$ for a training set of size $n$ will not exclude the minimizer of the true risk $w^* = \arg\min_{w \in W} R(\pi_w)$ in the policy space $W$ with probability at least*

$$1 - 2\exp\left(-2n\epsilon^2 p^2\right). \tag{30}$$

*Proof.* For the optimal $w^*$, let

$$S = \sum_{i=1}^n \frac{\pi_{w^*}(y_i \mid x_i)}{\pi_0(y_i \mid x_i)} \tag{31}$$

be the control variate in the denominator of the SNIPS estimator. $S$ is a random variable that is a sum of bounded random variables between 0 and

$$\max_{x,y} \frac{\pi_{w^*}(y \mid x)}{\pi_0(y \mid x)} \leq \frac{1}{p}. \tag{32}$$

We can bound the probability that the control variate $S$ of the optimum $w^*$ lies outside of $[1-\epsilon, 1+\epsilon]$ via Hoeffding's inequality:

$$P(|S - 1| \geq \epsilon) \quad \leq \quad 2\exp\left(\frac{-2n^2\epsilon^2}{n(1/p)^2}\right) \tag{33}$$

$$= \quad 2\exp\left(-2n\epsilon^2 p^2\right). \quad \square \tag{34}$$

The same argument applies to any individual policy $\pi_w$, not just $w^*$. Note, however, that it can still be highly likely that at least one policy $\pi_w$ with $w \in W$ shows a large deviation in the control variate for high-capacity $W$, which can lead to propensity overfitting when using the naive IPS estimator.

## C   APPENDIX: WHY DIRECT STOCHASTIC OPTIMIZATION OF RATIO ESTIMATORS IS NOT POSSIBLE.

Suppose we have a dataset of $n$ BLBF samples $D = \{(x_1, y_1, \delta_1, p_1) \ldots (x_n, y_n, \delta_n, p_n)\}$ where each instance is an i.i.d. sample from the data generating distribution. In the sequel we will be considering two datasets of $n+1$ samples, $D' = D \cup \{(x', y', \delta', p')\}$ and $D'' = D \cup \{(x'', y'', \delta'', p'')\}$ where $(x', y', \delta', p') \neq (x'', y'', \delta'', p'')$ and $(x', y', \delta', p'), (x'', y'', \delta'', p'') \notin D$.

For notational convenience, let $f_i \coloneqq \delta_i \frac{\pi_w(y_i|x_i)}{\pi_0(y_i|x_i)}$, and $\dot{f}_i \coloneqq \nabla_w f_i$; $g_i \coloneqq \frac{\pi_w(y_i|x_i)}{\pi_0(y_i|x_i)}$, and $\dot{g}_i \coloneqq \nabla_w g_i$.

First consider the vanilla IPS risk estimate of Eq. (5).

$$\hat{R}_{IPS}(\pi_w) = \frac{1}{n} \sum_{i=1}^{n} \delta_i \frac{\pi_w(y_i \mid x_i)}{\pi_0(y_i \mid x_i)} = \frac{1}{n} \sum_{i=1}^{n} f_i.$$

To maximize this estimate using stochastic optimization, we must construct an *unbiased gradient estimate*. That is, we randomly select one sample from $D$ and compute a gradient $\alpha((x_i, y_i, \delta_i, p_i))$ and we require that

$$\nabla_w \hat{R}_{IPS}(\pi_w) = \frac{1}{n} \sum_{i=1}^{n} \dot{f}_i = \mathbb{E}_{i \sim D} \left[ \alpha((x_i, y_i, \delta_i, p_i)) \right].$$

Here the expectation is over our random choice of $1$ out of $n$ samples. Observe that $\alpha((x_i, y_i, \delta_i, p_i)) = \dot{f}_i$ suffices (and indeed, this corresponds to vanilla SGD):

$$\mathbb{E}_{i \sim D} \left[ \alpha((x_i, y_i, \delta_i, p_i)) \right] = \sum_{i=1}^{n} \frac{1}{n} \alpha((x_i, y_i, \delta_i, p_i)) = \frac{1}{n} \sum_{i=1}^{n} \dot{f}_i = \nabla_w \hat{R}_{IPS}(\pi_w).$$

Other choices of $\alpha(\cdot)$ can also produce unbiased gradient estimates, and this leads to the study of stochastic variance-reduced gradient optimization.

Now let us attempt to construct an unbiased gradient estimate for Eq. (8):

$$\hat{R}_{SNIPS}(\pi_w) = \frac{\sum_{i=1}^{n} f_i}{\sum_{i=1}^{n} g_i}.$$

Suppose such a gradient estimate exists, $\beta((x_i, y_i, \delta_i, p_i))$. Then,

$$\nabla_w \hat{R}_{SNIPS}(\pi_w) = \nabla_w \frac{\sum_{i=1}^{n} f_i}{\sum_{i=1}^{n} g_i} = \mathbb{E}_{i \sim D} \left[ \beta((x_i, y_i, \delta_i, p_i)) \right] = \frac{1}{n} \sum_{i=1}^{n} \beta((x_i, y_i, \delta_i, p_i)).$$

This identity is true for any sample of BLBF instances – in particular, for $D'$ and $D''$:

$$\nabla_w \frac{\sum_{i=1}^{n} f_i + f'}{\sum_{i=1}^{n} g_i + g'} = \sum_{i=1}^{n} \frac{1}{n+1} \beta((x_i, y_i, \delta_i, p_i)) + \frac{\beta((x', y', \delta', p'))}{n+1},$$

$$\nabla_w \frac{\sum_{i=1}^{n} f_i + f''}{\sum_{i=1}^{n} g_i + g''} = \sum_{i=1}^{n} \frac{1}{n+1} \beta((x_i, y_i, \delta_i, p_i)) + \frac{\beta((x'', y'', \delta'', p''))}{n+1}.$$

Subtracting these two equations,

$$\nabla_w \left( \frac{\sum_{i=1}^{n} f_i + f'}{\sum_{i=1}^{n} g_i + g'} - \frac{\sum_{i=1}^{n} f_i + f''}{\sum_{i=1}^{n} g_i + g''} \right) = \frac{\beta((x', y', \delta', p')) - \beta((x'', y'', \delta'', p''))}{n+1}.$$

The LHS clearly depends on $\{(x_i, y_i, \delta_i, p_i)\}_{i=1}^{n}$ in general, while the RHS does not! This contradiction indicates that no construction of $\beta$ that only looks at a sub-sample of the data can yield an unbiased gradient estimate of $\hat{R}_{SNIPS}(\pi_w)$.

## D  APPENDIX: VARIANCE REGULARIZATION

Unlike in conventional supervised learning, a counterfactual empirical risk estimator like $\hat{R}_{IPS}(\pi_w)$ can have vastly different variances $\mathrm{Var}(\hat{R}_{IPS}(\pi_w))$ for different $\pi_w$ in the hypothesis space (and $\hat{R}_{SNIPS}(\pi_w)$ as well) (Swaminathan & Joachims, 2015b). Intuitively, the "closer" the particular $\pi_w$ is to the exploration policy $\pi_0$, the larger the effective sample size (Owen, 2013) will be and the smaller the variance of the empirical risk estimate. For the optimization problems we solve in Eq. (18), this means that we should trust the $\lambda$-translated risk estimate $\hat{R}_{IPS}^{\lambda_j}(\pi_w)$ more for some $w$ than for others, as we use $\hat{R}_{IPS}^{\lambda_j}(\pi_w)$ only as a proxy for finding the policy that minimizes its expected value (i.e., the true loss). To this effect, generalization error bounds that account for this variance difference (Swaminathan & Joachims, 2015b) motivate a new type of overfitting control. This leads to the following training objective (Swaminathan & Joachims, 2015b), which can be thought of as a more reliable version of Eq. (18):

$$\hat{w}_j = \arg\min_{w} \left[ \hat{R}_{IPS}^{\lambda_j}(\pi_w) + \gamma \sqrt{\frac{\widehat{\mathrm{Var}(\hat{R}_{IPS}^{\lambda_j}(\pi_w))}}{n}} \right]. \tag{35}$$

Here, $\widehat{\mathrm{Var}(\hat{R}_{IPS}^{\lambda_j}(\pi_w))}$ is the estimated variance of $\hat{R}_{IPS}^{\lambda_j}(\pi_w)$ on the training data, and $\gamma$ is a regularization constant to be selected via cross validation. The intuition behind this objective is that we optimize the upper confidence interval, which depends on the variance of the risk estimate for each $\pi_w$. While this objective again does not permit SGD optimization in its given form, it has been shown that a Taylor-majorization can be used to successively upper bound the objective in Eq. (35), and that typically a small number of iterations suffices to converge to a local optimum (Swaminathan & Joachims, 2015b). Each such Taylor-majorization is again of a form

$$\frac{1}{n}\sum_{i=1}^{n}\left[ A\left( \frac{\pi_w(y_i \mid x_i)}{\pi_0(y_i \mid x_i)} \right) + B\left( \frac{\pi_w(y_i \mid x_i)}{\pi_0(y_i \mid x_i)} \right)^2 \right] \tag{36}$$

for easily computable constants $A$ and $B$ (Swaminathan & Joachims, 2015b), which allows for SGD optimization.

