# OpenReview forum: "Deep Learning with Logged Bandit Feedback"
_ICLR.cc/2018/Conference — Accept (Poster)_

### Official Review · AnonReviewer1 · 2017-11-26
**Straighforward extension of existing techniques. Nevertheless, this is important work and I can see it being used by others.**

**Rating:** 7
**Confidence:** 4

**Review:**

Learning better policies from logged bandit feedback is a very important problem, with wide applications in internet, e-commerce and anywhere it is possible to incorporate controlled exploration. The authors study the problem of learning the best policy from logged bandit data. While this is not a brand new problem, the important and relevant contribution that the authors make is to do this using policies that can be learnt via neural networks. The authors are motivated by two main applications: (i) multi-class classification problems with bandit feedback (ii) ad placements problem in the contextual bandit setting.

The main contributions of the authors is to design an output layer that allows training on logged bandit feedback data. Traditionally in the full feedback setting (setting where one gets to see the actual label and not just if our prediction is correct or incorrect) one uses cross-entropy loss function to optimize the parameters of a deep neural network. This does not work in a bandit setting, and previous work has developed various methods such as inverse-propensity scoring, self-normalized inverse propensity scoring, doubly robust estimators to handle the bandit setting. The authors in this paper work with self-normalized inverse propensity scoring as the technique to deal with bandit feedback data. the self normalized inverse propensity estimator (SNIPS) that the authors use is not a new estimator and has been previously studied in the work of Adith Swaminathan and co-authors. However, this estimator being a ratio is not an easy optimization problem to work with. The authors use a fairly standard reduction of converting ratio problems to a series of constrained optimization problems. This conversion of ratio problems to a series of constrained optimization problems is a standard textbook problem, and therefore not new. But, i like the authors handling of the constrained optimization problems via the use of Lagrangian constraints. It would have been great if the authors connected this to the REINFORCE algorithm of Williams. Unfortunately, the authors do not do a great job in establishing this connection, and I hope they do this in the full version of the paper.  The experimental results are fairly convincing and i really do not have any major comments. Here are my "minor" comments.

1. It would be great if the authors can establish connections to the REINFORCE algorithm in a more elaborate manner. It would be really instructive to the reader.

2.  On page 6,  the authors talk about lowering the learning rate and the learning rate schedule. I am guessing this is because of the intrinsic high variance of the problem. It would be great if the authors can explain in more detail why they did so.

---

> ### Author Response · Authors · 2017-12-23
> **Author Response**
>
> Thank you for the comments. We will follow your suggestion and elaborate on the connection to REINFORCE in the on-policy setting as outlined in the other response. The reason why we reduced the learning rate is two-fold. First, we did observe convergence issues at the setting used for cross-entropy training, and agree hat this is probably due to increased variance. In addition, note that cross-entropy and our ERM objective are simply quite different and produce gradients at different scale. More generally, there is probably more room for improving convergence speed, like the use of alternative minibatch sizes, but this is besides the main point of this paper.

---

### Official Review · AnonReviewer3 · 2017-11-27
**The paper proposes a counterfactual risk minimization objective to perform learning from bandit feedback using a deep neural network.**

**Rating:** 8
**Confidence:** 3

**Review:**

In this paper, the authors propose a new output layer for deep networks allowing training on logged contextual bandit feedback. They propose a counterfactual risk minimization objective which makes the training procedure different from the one that uses conventional cross-entropy in supervised learning. The authors claim that this is the first attempt where Batch Learning from Bandit Feedback (BLBF) is performed using deep learning.

The authors demonstrate the derivation steps of their theoretical model and present 2 empirical results. The first result is on visual object classification using CIFAR-10.  To simulate logged bandit feedback for CIFAR-10, the authors perform the standard supervised to bandit conversion using a hand-coded logging policy that achieves 49 % error rate on training data. Using the logged bandit feedback data for the proposed bandit model, the authors are able to achieve substantial improvement (13% error rate) and given more bandit feedback, the model is able to compete with the same architecture trained on the full-information using cross entropy (achieving 8.2% error rate).

The second result is a real-world verification of the proposed approach (as the logged feedback are real and not synthesized using a conversion approach) which is an advertisement placing task from Criteo’s display advertising system (Lefortier et al., 2016). The task consists of choosing the product to display in the ad in order to maximize the number of clicks. The proposed deep learning approach improve substantially on state-of-the-art. The authors show empirically that the proposed approach is able to have substantial gain compared to other methods. The analysis is done by performing ablation studies on context features which are not effective on linear models.

The paper is well written. The authors make sure to give the general idea of their approach and its motivation, detail the related work and position their proposed approach with respect to it. The authors also propose a new interpretation of the baseline in “REINFORCE-like” methods where it makes the counterfactual objective equivariant (besides the variance reduction role). The authors explain their choice of using importance sampling to estimate the counterfactual risk. They also detail the arguments for using the SNIPS estimator, the mathematical derivation for the training algorithm and finally present the empirical results.

Besides the fact that empirical results are impressive, the presented approach allows to train use deep nets when manually labelling full-information feedback is not viable.

In section 4.2, the neural network that has been used is a 2 layer network with tanh activation. It is clear that the intention of the authors is to show that even with a simple neural architecture, the gain is substantial compared to the baseline method, which is indeed the right approach to go with. Still, it would have been of a great benefit if they had used a deeper architecture using ReLU-based activations.

---

> ### Author Response · Authors · 2017-12-23
> **Author Response**
>
> Thank you for the comments and the suggestion. We are planning to work with Criteo on further improving the results, and we agree that other architectures may perform substantially better. However, the key point of this paper is exploring the properties of our approach, not necessarily squeezing the last bit of performance out of any particular dataset. Note that the CIFAR10 results using the ResNet architecture already demonstrate that training deep and complex models using our approach is possible.

---

### Official Review · AnonReviewer2 · 2017-11-28
**A decent paper with some issues**

**Rating:** 6
**Confidence:** 3

**Review:**

This paper proposes a new output layer in neural networks, which allows them to use logged contextual bandit feedback for training. The paper is well written and well structured.


General feedback:

I would say the problem addressed concerns stochastic learning in general, not just SGD for training neural nets. And it's not a "new output layer", but just a softmax output layer (Eq. 1) with an IPS+baseline training objective (Eq. 16).


Others:

- The baseline in REINFORCE (Williams'92), which is equivalent to introduced Lagrange multiplier, is well known and well defined as control variate in Monte Carlo simulation, certainly not an "ad-hoc heuristic" as claimed in the paper [see Greensmith et al. (2004). Variance Reduction for Gradient Estimates in Reinforcement Learning, JMLR 5.]

- Bandit to supervised conversion: please add a supervised baseline system trained just on instances with top feedbacks -- this should be a much more interesting and relevant strong baseline. There are multiple indications that this bandit-to-supervised baseline is hard to outperform in a number of important applications.

- The final objective IPS^lambda is identical to IPS with a translated loss and thus re-introduces problems of IPS in exactly the same form that the article claims to address, namely:
    * the estimate is not bounded by the range of delta
    * the importance sampling ratios can be large; samples with high such ratios lead to larger gradients thus dominating the updates. The control variate of the SNIPS objective can be seen as defining a probability distribution over the log, thus ensuring that for each sample that sample’s delta is multiplied by a value in [0,1] and not by a large importance sampling ratio.
    * IPS^lambda introduces a grid search which takes more time and the best value for lambda might not even be tested. How do you deal with it?

- As author note, IPS^lambda is very similar to an RL-baseline, so results of using IPS with it should be reported as well:
    In more detail, Note:
    1. IPS for losses<0 and risk minimization: raise the probability of every sample in the log irrespective of the loss itself
    2. IPS for losses>0 and risk minimization: lower the same probability
    3. IPS^lambda: by the translation of the loss, it divides the log into 2 groups: a group whose probabilities will be lowered and a group whose probabilities will be raised (and a third group for delta=lambda but the objective will be agnostic to these)
    4. IPS with a baseline would do something similar but changes over time, which means the above groups are not fixed and might work better. Furthermore, there is no hyperparameter/grid search required for the simple RL-baseline
    -> results of using IPS with the RL-baseline should be reported for the BanditNet rows in Table 1 and in CIFAR-10 experiments.

- What is the feedback in the CIFAR-10 experiments? Assuming it's from [0..1], and given the tested range of lambdas, you should run into the same problems with IPS and its degenerate solutions for lambdas >=1.0. In general, how are your methods behaving for lambda* (corresponding to S*) such that makes all difference (delta_i - lambda*) positive or negative?

- The claim of Theorem 2 in appendix B does not follow from its proof: what is proven is that the value of S(w) lies in an interval [1-e..1+e] with a certain probability for all w. It says nothing about a solution of an optimization problem of the form f(w)/S(w) or its constrained version. Actually, the proof never makes any connection to optimization.

- What the appendix C basically claims is that it's not possible to get an unbiased estimate of a gradient for a certain class of non-convex ratios with a finite-sum structure. This would contradict some previously established convergence results for this type of problems: Reddi et al. (2016) Stochastic Variance Reduction for Nonconvex Optimization, ICML and Wang et al. 2013. Variance Reduction for Stochastic Gradient Optimization, NIPS. On the other hand, there seem to be no need to prove such a claim in the first claim, since the difficulty of performing self-normalized IPS on GPU should be evident, if one remembers that the normalization should run over the whole logged dataset (while only the current mini-batch is accessible to the GPU).

---

> ### Author Response · Authors · 2017-12-23
> **Author Response**
>
> Thank you for the detailed comments that will help us further improve the paper. We agree that we should clarify the connection and similarities to the REINFORCE baseline, and we will point out in more detail how the baseline is different in the off-policy setting we consider. First, we cannot sample new roll-outs from the current policy under consideration, which means we cannot use the standard variance-optimal baseline estimator used in REINFORCE. Second, we tried using the (estimated) expected loss of the learnt policy as the baseline as is commonly done in REINFORCE. As Figure 1 shows, it is between 0.130 and 0.083 (i.e. 0/1 loss on the test set) for the best policies we found. Figure 2 (left) shows that these baseline values are well outside of the optimum range of about [0.75-1.0]. Finally, the right way to modify adaptive baseline estimation procedures from REINFORCE to the off-policy setting remains an open question, and it is unclear whether gradient variance (as opposed to variance of the ERM objective) is really the key issue in batch learning from bandit feedback.
>
> To clarify your comment "final objective IPS^lambda is identical to IPS with a translated loss and thus re-introduces problems of IPS": Note that none of the individual IPS^lambda is used as an estimate, and the actual estimate we are optimizing in Equation (11) is bounded. While the SNIPS estimate can substantially reduce variance, large weights can certainly still be an issue that cannot be overcome without additional side information. And while the grid search increases training time, the empirical results (especially Figure 2) show that the prediction performance is not particularly sensitive to the exact value of lambda.
>
> Regarding your comment on "degenerate solutions for lambdas >= 1": The solutions are not necessarily degenerate, but they are suboptimal. This is shown in Figure 2.
>
> Regarding your comment on Theorem 2: You are absolutely right, and thank you for spotting this. We should be referring to the minimizer of the true risk in the statement of the theorem, not the minimizer of the empirical risk. What the theorem is supposed to say is: limiting the search to that range is unlikely to exclude the minimizer of the true risk. We will fix this in the final version.
>
> Regarding your comment on a "baseline system trained just on instances with top feedbacks": We are not sure what you mean by this baseline. One possible interpretation is: collect all unique contexts, and pair each context with action that has the highest observed reward in the logged dataset. Train on this manufactured dataset using supervised learning approaches. However, we generally don't assume that we repeatedly see that same context multiple times, so it is not clear that this is really a practical baseline.
>
> Regarding your comment on Appendix C: Our result does not contradict recent results on mini-batch optimization of finite sums of non-convex functions \sum_i f_i. While the SNIPS objective can be written as a finite-sum of non-convex f_i, mini-batch {f_i} in these problems do not correspond to our mini-batches {delta_i, impwt_i}. Since GPU can only hold a mini-batch and the normalizer requires the entire dataset, we may be tempted to explore ways of estimating the normalizer sufficiently well using mini-batches. Appendix C shows that any such approach is always going to give a biased gradient, justifying the effort in developing the Lagrangian approach instead.

---

### Public Comment · ~Charles_Elkan1 · 2017-12-02
**how to learn a better policy given batch data acquired using an old policy**

This is a good paper. What it shows fundamentally is how to learn a better policy given batch data acquired using an old policy. There are many applications for this in industry, but what people do usually is hand-craft a new policy that might be better, then do a real-world AB test. That is a slow and expensive process.

The big practical obstacle to using the method of this paper, or of related papers, is that current production systems usually haven’t recorded probabilities for actions taken by the old policy. More fundamentally, existing production policies are usually partly deterministic, so these probabilities don’t exist in a meaningful way.

Sections 1 to 3.1 are a good introduction and should be read even if one doesn’t have time for the later details. One quibble is that “equivariant” should be defined and discussed earlier than the first paragraph of 3.2.

Being more explicit about intuitions would be useful. Roughly, when an added constant shifts losses to be positive numbers, policies that put as little probability mass as possible on the observed actions have low risk estimates. If the constant shifts losses to the negative range, the opposite is the case. For either choice, the new policy eventually selected by the learning algorithm can be dominated by where the historical policy happens to sample data, not by which actions have low loss.

A useful note in the paper (Eqn 1) is that for a neural net to define a probabilistic policy, all that is needed is probabilities at the output layer, such as with a softmax. More fancy Bayesian methods are not needed. This is a valuable simplifier in practice and in theory.

3.3 should get to the point more quickly. The actual algorithm is simply to try Eqn 16 for several different values of lambda in a range below and above 1.0. How to then choose the best lambda could be explained more clearly. Also, how does the range change when the scale of delta changes?

CIFAR-10 has 60,000 labeled images with ten classes, so one could say there are 600K binary labels available. Figure 1 shows that using 220K of these is enough. It is not clear whether the better accuracy with 250K is a genuine phenomenon; if so, it needs explanation and to continue the experiment above 250K.

Criteo results are strong. Question: What is the ceiling on this dataset? How close does the new algorithm get to the best possible?

---

> ### Author Response · Authors · 2017-12-23
> **Author Response**
>
> Thank you for the comments and the suggestions regarding the presentation of the results. This is very helpful and we will work them in.
>
> Regarding your observation that the performance of BanditNet slightly dips below the supervised method, we were intrigued by this ourselves and have followed up on this. While the dip in Figure 1 is not significant, other experiments have shown that optimizing an ERM objective instead of cross-entropy does seem to produce a small but consistent advantage in prediction error. We are planning to explore this further, but this is outside the scope of this paper.
>
> The ceiling on the Criteo data is not known. But note that the logging policy is an actual production policy that Criteo is using. However, one has to keep in mind that Criteo may not be optimizing clicks alone, but that they may also consider other business metrics.

---

### Author Response · Authors · 2018-01-05
**Revision**

We thank everybody again for their useful suggestions and we uploaded a revision of the paper. The main changes in the revision are as follows:

- We clarified the relation to the REINFORCE baseline as part of the related work in Section 2, a more detailed paragraph at the end of Section 3.3, and a discussion of using the expected loss as a heuristic as part of the empirical results in Section 4.
- We fixed the statement of Theorem 2 in Appendix B and added some explanation about the intuition behind it.
- The equivariance problem as it applies to loss translations is now more formally defined in Section 3.2. We also added an intuitive explanation of what equivariance means in this context already in the introduction of the paper.
- Since the co-author who conducted the Criteo experiments became unresponsive since submission of the paper and was removed from the author list, the remaining authors felt uncomfortable including the results and vouching for their correctness. We therefore removed them from the paper, since the main points about training deep networks with logged bandit feedback are already made by the ResNet experiments.

---

### Decision · Program_Chairs · 2018-01-29
**ICLR 2018 Conference Acceptance Decision**

**Decision:**

Accept (Poster)

**Comment:**

In this paper the authors show how to allow deep neural network training on logged contextual bandit feedback. The newly introduced framework comprises a new kind of output layer and an associated training procedure. This is a solid piece of work and a significant contribution to the literature, opening up the way for applications of deep neural networks when losses based on manual feedback and labels is not possible.